# Pushing for Structural Reforms: Impacts of Racism and Xenophobia upon the Health of South Asian Communities in Ontario, Canada

**DOI:** 10.3390/ijerph22111668

**Published:** 2025-11-03

**Authors:** Manvi Bhalla, Ève Dubé, Noni MacDonald, Helana Marie Boutros, Samantha B. Meyer

**Affiliations:** 1Institute for Resources, Environment and Sustainability, University of British Columbia, Vancouver Campus, AERL Building, 429-2202 Main Mall, Vancouver, BC V6T 1Z4, Canada; manvi.bhalla@ubc.ca; 2Department of Anthropology, Lavel University, 1030, Avenue des Sciences-Humaines Suite 3456, Quebec, QC G1V 0A6, Canada; eve.dube@ant.ulaval.ca; 3Department of Pediatrics, Dalhousie University, IWK Health Centre, 5850/5980 University Ave, Halifax, NS B3K 6R8, Canada; noni.macdonald@dal.ca; 4Department of Health, Aging, & Society McMaster University, 1280 Main Street West, Hamilton, ON L8S 4L8, Canada; boutrosh@mcmaster.ca; 5School of Public Health Sciences, University of Waterloo, 200 University Avenue West, Waterloo, ON N2L 3G1, Canada

**Keywords:** qualitative, structural determinants, COVID-19, racism, xenophobia, stigma, South Asian, Ontario

## Abstract

South Asian (SA) communities in Ontario, Canada experienced disproportionately higher rates of COVID-19 infection. Moreover, these communities also faced racism fueled by COVID-19-related misinformation and xenophobic sentiments that placed blame on them for virus transmission. The aim of this research was to understand, from the perspective of local SA communities, the causes behind higher incidences of COVID-19. SA adults (*N* = 25) participated in a focus group (*N* = 3) investigating experiences during the early stages of the pandemic. Data, interpreted through the lens of the Public Health Critical Race Praxis, suggest that the structural determinants of health, alongside racism and xenophobia, negatively impacted health outcomes for these communities. By not taking an active anti-racist stance, media, health and government authorities were viewed as perpetuating discriminatory narratives and practices, fueling blame and stigma towards these South Asian communities for COVID-19 transmission. Local public health policies, practices and communications were perceived to be informed by, and best serve, white Anglo-European settlers. This research provides insight into the role that health officials can play in addressing local and regional discrimination and stigma to promote equity-centered disease prevention efforts. Our findings should be integral to current and ongoing research and action related to pandemic preparedness.

## 1. Introduction

The COVID-19 pandemic highlighted social and structural determinants of health inequities, with morbidity and mortality rates, globally impacting vulnerable populations [1]. Canada, the location of the present research, saw disparities in both exposure, disease prevalence and health outcomes. Central to the proposed research are data from very early stages of the COVID-19 pandemic. When Canadian public health agencies themselves were not collecting race-based data [2], members of South Asian communities were researching and advocating for public health structures to help address the disproportionately higher COVID-19 transmission rates felt by their communities [3]. Consequently, when these data began to be collected and findings emerged, it was confirmed that regions in Ontario, Canada with a higher density of South Asian residents reported disproportionately higher COVID-19 infection rates [4,5]. For example, within Toronto in 2020, the highest proportion of both COVID-19 cases and deaths were amongst South Asians [6]. To add important context, South Asians make up the largest racially-minoritized (i.e., non-white) group in Canada (7.1% of total population; 2.2 million people); and this is the same for the most populous province in the country, Ontario (10.8% of total population) [7]. These numbers are bolstered in great part by immigration, whereby over half of all immigrants to Canada (51.5%) originate from Asia, of which South Asians make up one of the largest proportions [8]. India is considered the leading country of birth for recent immigrants to Canada, with approximately one in five recent immigrants (18.6%) being born there [9]. Thus, South Asians, and new immigrants in particular, constitute an important population to consider in the context of ongoing public health discourse surrounding race and ethnicity-based health inequities in Canada.

Asian communities continue to experience undue racism in Canada, and this was fueled further by COVID-19-related misinformation and xenophobic sentiments, particularly in the early stages of the pandemic [10]. Miconi et al. [11] report from their survey of Quebec residents that non-white participants were at higher risk of experiencing COVID-19-related discrimination, with East Asian participants being four times more likely, followed by South Asian and South-East Asian participants. This discriminatory discourse has promoted significant increases in incidences of Asian community members experiencing physical and verbal assaults, being told to “go back to their country”, property damage, being coughed on and workplace discrimination; this includes South Asians and South-West Asian/North African individuals being more likely to experience job insecurity due to discrimination based on their immigration status [12]. These sentiments have also extended into the online sphere promoting cyber racism and xenophobia [13]. Underrepresented in the popular discourse on these issues, however, are the embodied experiences and realities of Ontario’s South Asian community members themselves.

### Theoretical/Conceptual Framework: Public Health Critical Race Praxis

Guo, Guo [14] postulate that from a Critical Race Theory perspective, Asian Canadian identities have been socially constructed to be seen as biologically inferior, culturally backward and racially undesirable. There are many historical examples of white settler-Canadians’ anti-immigration stances since the colonization of these lands; from the Chinese head tax and Chinese Exclusion Act to the forced relocation and internment of Japanese Canadians, to the 1914 Komagata Maru incident which where 376 Punjabi passengers sat in a ship in Vancouver harbor for two months, then were forced to return to India. Asian communities were not able to vote in elections until 1947, before which signs were put up stating, “will arrest Hindu who is alleged to have voted” (p. 21) [15]. Under this theoretical view, racism structures how society functions, and it is not an anomaly—therefore, this approach asks one to consider the socio-historical contexts that enable racism to operate. Importantly, it challenges the “production and reproduction of national narratives that construct Canada as an example of a ‘successful’ multicultural society” [16], given its ongoing assimilative policies and practices.

Given that Asian Canadians were not recognized as citizens less than three-quarters of a century ago, it is not surprising that systemic inequities persist into our present day and that they are having downstream impacts on all dimensions of life for these communities.

Generated by adapting tenets of Critical Race Theory to the public health context, the Public Health Critical Race Praxis (PHCRP) is an interdisciplinary framework that asks public health practitioners to look to the root causes of health inequities and critically investigate the role that racism and other systemic factors play in creating and shaping health outcomes, in lieu of settling upon individual determinants such as biological, behavioral and/or modifiable lifestyle factors alone [17,18]. The PHCRP’s iterative and action-oriented methodology has already been established to be useful in elucidating how structural racism contributes to the disproportionate burden of COVID-19 as experienced by communities of color [19,20]. The PHCRP postulates that racism is a structural problem that is maintained by intentional and/or unintentional actions that ‘vulnerabilize’ (or produce vulnerabilities in), and/or exploit existing vulnerabilities of, a subpopulation to cause harm to its health and livelihood [19]. A failure to name and address systemic discrimination as a root cause maintains a status quo that privileges certain groups’ health to the disadvantage of others [21]. Methodologically, the PHCRP deters researchers from employing “color blindness” or post-racial perspectives that obscure the lived realities experienced by communities of color today. Instead, it asks researchers to “center the margins” and offer opportunities for communities of color to speak for themselves in sharing their lived realities [22]. In doing so, it offers a critical opportunity to identify inequities that would persist on the basis of race-based discrimination alone, independent of one’s income, class and socioeconomic status, for example. Further, it also helps to capture the intersectional realities of individuals for whom differential and cumulative lived experiences of oppression have created compounding health inequities.

The objective of this research is to understand, from the perspective of local South Asian communities, the causes behind higher incidences of COVID-19 and to give voice to their experiences of stigma and discrimination during the COVID-19 pandemic. In doing so, we aim to document lessons learned that can be used to inform the changes needed in Public Health and society broadly if we are to guard against structural inequities in the case of future pandemics.

## 2. Materials and Methods

### 2.1. Study Population and Recruitment

In early 2021, recruitment posters in English were shared on Instagram and Twitter accounts created for the purpose of promoting this research study. They advertised our study focus and mentioned that we were recruiting South Asian people over 18 years of age residing in Ontario to participate in a focus group for CAD 20 in remuneration for their time. Focus groups were advertised as being held over Zoom or Webex, due to the ongoing pandemic, and consisted of 40–60-minute-long sessions with 8–10 other individuals. Interested parties were asked to email the research team directly to express their interest. Upon receipt of emails, our team provided prospective participants with a link to a demographic survey hosted on Qualtrics to complete to determine their eligibility based on their self-identification of being South Asian. This survey also collected their age, gender, racial/ethnic background, marital status, highest level of institutional education completed, occupation and city or town they reside in within the province of Ontario. Eligible participants were provided with consent forms, and those who enrolled in the study were provided with their virtual focus group time, date and link.

### 2.2. Study Sample

#### 2.2.1. Demographic Characteristics

A total of 25 individuals participated in this study. They self-identified as Sri Lankan (*n* = 2), Tamil (*n* = 1), Pakistani (*n* = 4), Punjabi (*n* = 2), Indian (*n* = 10), mixed East African/Indian and Pakistani (*n* = 1) or South Asian broadly (*n* = 5). There were 20 women, 5 men and no non-binary individuals. Participants’ ages ranged from 19 to 64 years old, with 72% of the sample being under 29 years of age (*n* = 18). In terms of marital status, 60% of the sample was single (*n* = 15), with some individuals being in a relationship (*n* = 3), living common law (*n* = 1) and married (*n* = 5). For the highest level of institutional education attained, 84% of the participants (*n* = 21) indicated that they had at least some university education. Notably, participants ranged across all of the levels indicated, including high school diploma (*n* = 1), some college education (*n* = 2), college certificate, diploma or degree (*n* = 1), some university education (*n* = 5), undergraduate degree (*n* = 6), master’s degree (*n* = 7), doctorate (*n* = 2) and professional degree (*n* = 1). Professionally, there was a great degree of diversity in the occupations of participants, including, broadly, students (*n* = 9), researchers, consultants and analysts (*n* = 5), healthcare workers (*n* = 3), academic institutional instructors/staff (*n* = 2), business professional (*n* = 2), public service (*n* = 1), food service (*n* = 1) and unemployed/not specified (*n* = 2).

#### 2.2.2. Geographic Context

Over 50% of participants (*n* = 13) resided within the health jurisdiction of the Peel Region and City of Toronto Health Units; these areas were known to be COVID-19 “hotspots”. Overall, study participants resided in 12 different cities/towns across Ontario, representing constituents for 10 different Public Health Units and all of Ontario’s Health Regions except for the northeast and northwest Health Regions (Figure 1).

### 2.3. Data Collection

In April 2021, semi-structured focus groups were facilitated by a young South Asian woman researcher who could converse in Hindi, Punjabi and Urdu via the online video platforms Webex and Zoom. Focus group questions were designed to understand experiences of COVID-19 to inform public health communication strategies. The interview guide used was adapted from a template generated from the wider research team collaborating on a Canadian Institutes of Health Research (CIHR) grant aiming to understand COVID-19 countermeasures acceptance to inform public health communication strategies across the country. Participants were encouraged to both participate verbally in the group and to type inputs in response to others actively via the chat feature. Both sources of data were recorded, and the audio files were transcribed verbatim (first transcribed using Otter.ai, then hand checked for accuracy and anonymized). The audio transcripts were then integrated with the real-time chat inputs to be analyzed as one continuous transcript, as per the novel approach we describe in our previously published paper on this topic [23].

### 2.4. Data Analysis

First, two members of the research team each independently coded one of three focus group transcripts in a Microsoft Word document using inductive line-by-line open coding, then focused coding [24]. Percent agreement between coders was determined to be 84%, which exceeded the interrater reliability standard set by Miles and Huberman [25]. Similarities and differences in approaches to coding and emergent themes were discussed before all three transcripts were subject to focused coding using NVivo by the lead researcher. A total of 8 themes and 44 sub-themes were generated. Five of these themes were discussed in a prior publication [23], and the three remaining themes are explored and discussed in this paper. Data were interpreted through the lens of the Public Health Critical Race Praxis [1,19,21]. All quotes shared in this paper are de-identified and use culturally appropriate pseudonyms for the participants, and their respective ages are provided in brackets.

### 2.5. Rigor

There were numerous actions taken to ensure rigor and a high quality of research throughout the design of this study. In addition to inter-rater reliability, as aforementioned, we employed expert checking with public health researchers on our national research project team, peer debriefing, use of direct quotes from participants in supporting our interpretation and analysis, and we stopped focus groups when we felt we had reached data saturation for our research topic, as many individuals were repeating inputs from other focus group sessions. We have also described our methods in great detail to support others in reproducing our research. Although we did not formally conduct member checking, we did strive to contextualize our findings within the wider public discourse implicating South Asian communities, namely in the media/press, at the time. Given that we were conducting this research in the context of the early stages of the COVID-19 pandemic where new information was always emerging, we did not formally employ triangulation in our research design.

### 2.6. Researcher Reflexivity/Positionality

Lead author MB, who identifies as a South Asian woman residing in the Greater Toronto and Hamilton Area (an area where disproportionate rates of COVID-19 were documented among SA communities) is experienced in conducting community-based research using qualitative methods. MB’s positionality and experience allowed for an ability to connect with participants linguistically and culturally during focus groups, allowing for a safer space for participants to share honest and elaborate responses. For example, participants moved between Hindi and English comfortably given their knowledge of MB’s ability to communicate in both languages. However, during data analysis and in consideration of bias, all audio recordings of focus groups, transcripts, coding, data interpretation and writing up of findings were discussed and co-generated with investigator SBM, who identifies as a white woman and who is an established public health scholar and expert in health risk communication, health promotion and qualitative health research methods. HMB, who identifies as a Coptic woman, also supported rigor in data analysis as a second coder for this research whilst completing her undergraduate degree. Serving as principal investigators in the national research project team behind the present research were applied public health researcher ED and doctor/scholar NM, who both identify as white women (located in Quebec and Nova Scotia, respectively) and also provided expert input regarding data analysis.

### 2.7. Ethics Statement

Informed consent was secured digitally by first providing all eligible participants with the consent form, then by reiterating the consent information at the start of each focus group session prior to beginning the recording. All procedures performed in studies involving human participants were in accordance with the ethical standards of the University of Waterloo’s Research Ethics Office (#42160) and with the 1964 Helsinki Declaration and its later amendments or comparable ethical standards.

## 3. Results

Participants across all focus groups expressed that the media, public health authorities and other government officials had perpetuated racism against South Asian communities during the course of the COVID-19 pandemic in overt and in direct ways. These entities were perceived to have placed blame for the community’s high case load to be a result of the community’s actions or inactions, rather than acknowledging the role of systemic oppression and the underservice of this community through policies, practices and communications designed without their input, compounded with the social determinants of health, namely income. Emergent themes connected the higher incidence of COVID-19 experienced by South Asian communities with specific instances of stigma and discrimination they were subject to from government and public health authorities. They viewed this stigma and discrimination as a root cause for underservice by these structures. Overall, participants expressed beliefs that South Asian patients often receive sub-standard care by health professionals, particularly when there are perceived cultural or language barriers, as compared to their white counterparts. Compounding this with other systemic inequities, such as employment in work with higher risk of SARS-CoV-2 exposure and discrimination in accessing higher paying, safer jobs despite qualifications, the role of systemic oppression in the prosperity of this community’s health emerged as a key finding.

### 3.1. Systemic Failures That Increased COVID-19 Risk and Transmission for South Asian Communities

Participants elaborated upon the compounding impacts of racism, income inequity, socioeconomic status and class upon the communities’ experience of COVID-19. For example, Gurleen (20), shared: “many South Asians are front line workers to support their families… They are at a higher risk because there may not be the same option for them to work from home.” Others added that South Asians are disproportionately forced to work low-income jobs, often due to no other choice. Amina (21) explains, “they do have to work those jobs like, you know, being an Uber driver or working in a restaurant because sometimes their degrees don’t transfer over here, so they can’t get the quality job that they had before.” Participants expressed that this income-based precarity does not exist in a vacuum but, rather, is an intersectional compounding reality of most Brown immigrants in Canada. Emotions ran high during many of these conversations; as Jai (51) illustrated, “[when] you [are a] white person—[and] I’m not being racist— eating the food on your dinner table, it’s easy to blame the South Asians as, ‘oh they are the ones spreading [it] around’ but remember who packed that food for you. Food packaging people, standing in lines. You know, they don’t have the luxury of having a 6m or 6-foot distance. It’s difficult to pinpoint and say South Asians [are the biggest spreaders], [or] as people coming from India spreaders. White people need to understand that [these people] don’t have [the] luxury [of choice] with [whole] households working, you know… and I’ve been through that when I came to this country.”

Participants also felt that despite knowledge of front-line worker precarity, public health structures failed to use harm reduction approaches to mitigate the compounding risks, particularly for those living in the ‘hot spots’ of Peel, Toronto and York Regions. Simrun (26), explained, “majority of the workers in Peel work with temporary agencies, in factories, warehouses, and precarious work. Most of them have unpaid sick leave as well so they’re conflicted between going to work while sick versus not receiving any pay… many are front line workers [with] unpaid sick leave [who have] multigenerational families in one house, gatherings between families, [and a] delayed prioritization of vaccines [despite their] high risk.” Many participants spoke to working at crowded factories such as the Amazon distribution centers or to taking up work in the ‘gig’ economy (e.g., Uber Easts), then returning to their multi-generational homes where they have few options in terms of isolating to minimize chances of their family’s exposure to the virus. Mohammad (29) felt that “blue-collar workers and people, like working class communities… have completely been abandoned, um, by this government, and also like, [it is an] institutional failure at almost every level… We’ve gone through so many aspects of failures of communication. Whether it’s like the shaming of people for not abiding by public health measures— which in the beginning, everyone was on board, but they’re like, ‘yes shame these people.’ But now, it’s like 14 months in and you’re still employing the same kinds of like naming and shaming tactics, and like, failing to protect people and failing to… create safer ways for people to interact with one another.” Overall, participants expressed that public health officials failed to use a harm reduction approach to mitigate the compounding risks experienced by “blue-collar working class” South Asian breadwinners, an experience that they express does not exist in the vacuum of income inequality but rather is an intersectional compounding reality of most Brown immigrants in Canada.

### 3.2. Discrimination Against South Asian Cultures, Religions and Ways of Being Fueled Blame and Stigma

Participants felt that the aforementioned oversights on the side of government officials and health authorities contributed to increased stigmatization of South Asian culture, holidays, religious practices and other ways of being. Elizabeth (36) shared, “I have to say it infuriates me that there’s undue blaming to this South Asian population in terms of weddings and get-togethers because first of all, I think it’s very there’s a long history of white power structures, blaming immigrants and blaming people of color and I think that we’re seeing that. We’re seeing like, there are people who are not complying with COVID-19 rules across all the kind of ethnic backgrounds and non-ethnic backgrounds in Canada. Like, we’ve seen parties broken up in Oakville and that you don’t see people writing the news, ‘White people are spreading COVID’ and you know, kind of blaming going the other way because, unfortunately, that’s not how the power structure works. So um, it really makes me angry. It makes me—it draws on a lot of things, like historical methods of shifting blame, and I don’t think it’s been properly rectified in like the popular news media at all.”

Moreover, white, Eurocentric policymaking was frequently referred to as being the standard, as many participants across all focus groups cited the example of the treatment of Diwali by officials and the media, as compared to Christmas. One woman, Deepika (20), even sarcastically noted that the stay-at-home order in December 2020 was only set up after boxing day. Similarly, Keerat (23) expressed, “back in October or November when it was Diwali, there was lots of stuff in the media about: ‘don’t see your family, stay home- this and that.’ Of course, I mean—*do that*, but I just think it’s funny that during Christmas time it was okay. If you have to see your family, wear a mask—[but] there was the option to see your family. But for us… [Diwali] is such an important and exciting time of the year… and it’s kind of like… really interesting the different ways that was treated in the media, especially in comparison to Christmas and I guess other, more traditional like white people-celebrated holidays.” Participants drew links between these examples of underservice of the population towards an apprehension in trusting the public health authorities or government. Iqbal (21) said, “that’s like a very structural… There’s a reason these things are happening. It’s because we *value* certain things in our society over you know, protecting people with paid sick days and things like that. It’s easy for the narrative to be like, ‘oh, it’s weddings or gatherings,’ when you know, of course, they play a part, but we know statistically that’s just straight up not the case.”

### 3.3. Racist and Xenophobic Narratives of South Asians as Sources of Mass COVID-19 Transmission

Across all focus groups, numerous participants expressed that the Premier of Ontario, among other government and public health officials, spread xenophobic sentiments towards immigrants of color by painting them as responsible for bringing emerging strains and variants of concern into the country. Many agreed that this further contributes to the “othering” of South Asian communities from the general population and has not only negatively impacted small business owners but also contributed towards a sense of mistrust in authority figures within Ontario and Canada. Two specific examples emerged in one focus group, when Jai (51) first recalled their perspective on an article they read in the Toronto Sun, stating it said that “the planes coming in from India are carrying *their* new variants… I distinctly remember that it said, ‘plane load’… COVID positive people in India [are] just lining up to get into aircrafts to come to Canada to spread it. [The reality is that] it was not plane loads of people coming into Canada to spread COVID. You know, yes, we have [an] outbreak [in India] but doesn’t mean that we’re coming specifically to spread COVID. It negates so much [about] us as a community, [and] you know it’s wrong and, in the end…the way the media is painting it, like, it’s more like a blame game, which definitely you know, [is a] collective punishment.”

Following up on this, Mohammad (29) singled out an opinion piece published in The Toronto Star as another example of the more widespread media bias placing blame on South Asian communities’ religious practices, cultural celebrations and other ways of being as solely responsible for their higher COVID-19 transmission rates. Released in November 2020, this piece, entitled, “South Asians play a part in COVID-19 transmission, and we need to acknowledge it”, was co-authored by a team of South Asian doctors and, as a result, had widely been shared [26,27]. It was consequently critiqued by numerous leading South Asian health researchers and community advocates [28,29,30,31]. Alongside sharing the article’s link in the chat during his focus group, Mohammad verbally shared their perspective on this kind of media coverage with the group, stating “I think that one of the most dangerous pieces of arguments that I heard was in this article from The Star. I was actually pretty disappointed in the fact that Star could publish something like this, but then again, I wasn’t really shocked… When you have a paper like The Star publishing like this article, which is coming from South Asian doctors, it adds a huge weight of legitimacy to it. The arguments that they’re presenting are saying that our values and our social way of interacting is what’s causing the spread of COVID-19. Never mind the fact that [they did not do a] class-based analysis. You’re putting people between a rock and a hard place where they have to go to work. But then, this article says like: ‘Oh, it’s our values that are internalized, like traditions, that somehow are responsible for us spreading COVID’. It’s blaming the victim. It’s so, so ridiculous and dumb. I mean, thankfully, there was a lot of backlash against this article as well.” A few other participants nodded and expressed agreement towards the harmful nature of such media coverage.

## 4. Discussion

In this study, we drew on data collected during the COVID-19 pandemic to better understand the social and structural factors that shape health inequities in SA communities to inform the changes needed in Public Health and society broadly if we are to guard against structural inequities in the case of future pandemics. Our findings suggest that while discrimination is perceived as fueling blame and stigma associated with COVID-19 transmission, structural factors and systemic oppression act as determinants implicating health outcomes for the South Asian communities in Ontario more than other factors intrinsic to the communities, including their cultural and religious practices.

Participant responses spoke to the role of structural factors as the root causes for disproportionately higher COVID-19 rates and negative health outcomes for South Asian communities. Systemic factors are noted in the wider literature; for example, increased risk of transmission and severity of COVID-19 cases in non-white populations has been identified as being often due to a combination of socioeconomic factors, barriers to accessing healthcare, increased prevalence of co-morbidities, alongside precarious living and work conditions that synergize with legacies and present-day experiences with discrimination and stigma [32]. Aligned with observations shared by our participants, living in multi-generational homes and taking care of one’s elders in lieu of sending them to care facilities is common in South Asian cultures. Indeed, racially/ethnically minoritized peoples, particularly new immigrants, are more likely to work in essential services that require them to be public facing, work in crowded facilities and/or do not offer adequate paid sick leave, which, in the context of increased economic pressures and lack of job security, increases the likelihood of exposure and outbreaks [32]. Similarly, it cannot be dismissed from this analysis that South Asians make up a significant portion (28.72%) of the ‘essential’ workforce (e.g., grocery stores clerks), resulting in a greater risk of exposure to SARS-CoV-2 [33]. In the context of COVID-19 in Ontario, a study conducted in the Greater Toronto Area (GTA) between April to July 2020 found that unemployment, apartment living and having an essential worker in the household were all associated with decreased likelihood of adherence to public health preventative measures [34], which, based on our data, may relate to an inability to comply with measures due to living and working conditions. For example, this was expressed by most of the participants in this study, particularly in their identification of the number of new immigrants working in precarious, hazardous essential service jobs with no alternative options for their livelihoods. This is also supported by pre-pandemic trends, as Statistics Canada has reported that at least 61% of South Asian women and 43% of South Asian men have experienced discrimination at work or when applying for a job or promotion [35].

Participants also spoke to the role oppression in the disproportionate rates of COVID-19 in South Asian communities. Previous research aligns with our participants’ remarks [36,37,38], as racism and xenophobia are argued to create the aforementioned conditions for precarities to exist, resulting in a manufactured inequity where South Asian communities are forced to bear a disproportionately higher burden of COVID-19 risk and transmission. Findings from research capturing experiences of stigma and discrimination during the pandemic also support these themes [37,38,39]. With this context, it is easier see from where harmful generalizations and implicit biases originate about Asian cultural gatherings and multigenerational homes as sources of outbreaks.

Our findings suggest that media, health and government authorities perpetuated discriminatory narratives and practices, which fueled blame and stigma towards South Asian communities for COVID-19 transmission. Narratives that placed blame for high transmission rates on South Asian celebrations such as Diwali went largely unchallenged by public health and government officials. These narratives persisted despite data from Ontario’s COVID-19 Advisory Committee stating that events, ceremonies and religious services constituted substantially lower sources for outbreaks (3% in Toronto, 5% in Peel, 1% in York) as compared to congregate living and industrial settings combined (15% in Toronto; 30% in Peel, 26% in York) [40]. Instead, public health actors and government officials have been criticized for relying on white, Eurocentric ways of being as a ‘default’ to inform countermeasures for COVID-19 and failing to provide these communities with contextual harms reduction measures to safely exist with their lived realities. The absence of widespread accessible health risk communications for safe gatherings, paid sick leave and being considered a priority population within the vaccination rollout contributed towards perpetuating the disproportionate risk and transmission of COVID-19 for South Asian communities. Lack of deliberate actions to ensure the health of the high proportion of South Asians, particularly new immigrants, residing in these regions who make up a significant portion of the front-line workforce [32,33], contributed to the manufacturing of ‘hotspots’ in Toronto, Peel and York.

Participants also expressed great frustration with the mainstream narratives that placed blame for higher transmission rates upon actions or inactions by South Asians— as well as how these narratives remain unchallenged by many public health and government officials. Despite Canada being celebrated as a multicultural country with a successful model of immigration, critical race perspectives progress because the country possesses deep-rooted discrimination due to discursive racialization [14]. This is aligned with our findings, which further suggest that discrimination from government and officials and consequent underservice fostered of a culture of self-advocacy among South Asian communities to help mitigate COVID-19 risk. Importantly, the ostracization of racially and ethnically minoritized communities in what is currently Canada reflects the divide between whiteness and non-whiteness, which is fueled by stigma rooted in the misperception of Western cultures being seen as superior to Asian cultures [14]. Participants, for example, suggested that these assumptions often lead to generalizations surrounding cultural gatherings and multigenerational homes. This may be considered as “pathologizing culture” and failing to identify the root causes of transmission in Asian communities, according to Dr. Bains, the director of the South Asian Studies Institute at the University of the Fraser Valley [31].

Our research underscores the need for public health to move upstream, from investigating social to structural determinants of health, to mitigate ongoing harm but also that which might arise in the case of future pandemics. In public health, a lens of social determinants of health is routinely employed to help investigate health inequities associated with social factors such as education level, job security, access to safe housing and/or identity factors such as one’s racial, ethnic and gender identity. While this approach is well intentioned, one of its emergent critiques is that it often fails to identify the structural and systemic factors that are the root causes behind the distributional injustices observed. Further, in associating health inequities with social and identity factors alone, this approach can contribute to pathologizing narratives of marginalized identities as being inherent to experiencing health inequities rather than being engineered by unjust systems [41]. To address these issues, one can move towards using a more structural analysis, which instead looks at how systems such as white supremacy, settler colonialism, capitalism, patriarchy and ableism work synergistically to manufacture social hierarchies that privilege some to the detriment of others. Then, this can be seen as creating the conditions for systemic oppression and discrimination to exist and persist, which serve as the root causes for health inequities for racially, ethnically and gender-minoritized, disabled and low-income peoples. In other words, one goes from acknowledging that racially-minoritized newcomers to Canada often have lesser access to social supports with the ability to determine health outcomes, such as job security and safe housing, to instead seeing this issue as a structural problem manufactured by white supremacy and settler colonialism, which create and maintain the societal conditions that privilege white people and fluent English speakers towards access to these social supports, to the disadvantage of those that do not fall within those categories. Thus, acknowledging the role of the structural determinants of health helps minimize further perpetuation of harmful, stigmatizing perceptions of communities of color as a homogenized cohort of low-income and uneducated peoples who are unable to help themselves [41]. Further, this type of analysis tends to focus more on upstream investigation of what is needed to actualize complex solutions that can work to address the root cause of systemic and structural issues. For example, the social determinants of health help us to identify what we need to collect race-based data to be able to study differences in COVID-19 exposure, but does it go so far as to ask why we were perhaps not collecting it in the first place? Under what normative conditions was it assumed by public health structures that the health status and needs of racially and ethnically-minoritized peoples in Canada would be comparable to those of white European-descendent settlers? To answer this question, one needs to investigate the larger problem of white, Western-centric worldviews and practices dominating public health and policy-making environments. Problematically, it was only after race-based data began to be collected that the dire realities of the pandemic as expressed by the lived experiences of racialized communities in Ontario came to be “verified” from the perspectives of Western scientific practices [42]; equity-centered approaches began to take center stage in the public health response, but already this negatively impacted trust between these communities and public health structures as well as government institutions. Overall, giving voice to members of the South Asian communities identified the critical need for public health and government officials to take an active, anti-racist stance, which would begin to create upstream change within the systems and structures to ensure that all public health work is not intentionally nor more commonly unintentionally underserving communities of color.

We feel it important to note that there were many instances in Ontario where service providers, government, public health and community organizations expanded resources (e.g., funding) to meet the needs of marginalized populations. Many also sought specific involvement from affected communities to better serve those most in need (including SA communities but also other marginalized populations). For example, ‘Apna Health’ was created during the pandemic and continues to provide services that promote the South Asian community’s health and wellness (e.g., cancer screening, diabetes management, dementia support). While Apna is largely community funded, it works in collaboration with Punjabi Community Health Services (PCHS), who receive funding from the Ontario government. We, as Canadians, share pride in many of the actions taken to address the pandemic and realize this was an unprecedented situation and it was inevitable that mistakes would be made, with science evolving and a workforce that was largely lacking in resources and training for the provision of culturally safe services. Our intention with this paper is to highlight the structural barriers that are amendable to change and need to be addressed in advance of the next public health crisis.

### Limitations

The limitations of our study included an overrepresentation of younger adults, with most of our sample being under 29 years of age. This is largely due to our recruitment being in English and being advertised primarily on social media. However, even when younger participants expressed perspectives concerning media they had been exposed to, for example, discussing South Asian communities, many older individuals tended to agree with their analyses. Moreover, youth often shared realities of their parents working in precarious positions, as those individuals themselves often were not able to participate in the focus groups due to their lack of time and/or capacity. We recognize that gender and sex do play a significant role in shaping health outcomes as they relate to COVID-19. We largely discuss these implications in our other paper, which focuses on an intersectional analysis of our thematic findings, with a dedicated focus on the unique lived experiences of South Asian women and girls during COVID-19 [23]. Despite this, we acknowledge that there were no non-binary or gender-diverse identifying individuals in our study, so our findings are likely limited to cisgender, heteronormative contexts; further research is needed to elucidate the unique experiences of non-binary and other gender-diverse individuals.

## 5. Conclusions

The deep inequities surrounding COVID-19 risk for South Asian communities reflect shortcomings in addressing the underlying mechanisms or determinants of COVID-19 transmission for all Canadians. It is critical that we learn from the harms of our actions in anticipation of future pandemics and public health crises. Our findings show that South Asians in Ontario believed they were wrongly blamed for higher COVID-19 transmission due to cultural and religious practices, instead of structural factors including the determinants of health. Race-based data collection validates the inequitable burden of transmission experienced by racially and ethnically minoritized peoples. Our research adds context for observations to prevent interpretations perpetuating blame, stigmatization and discrimination now and moving forward. Public health policies, practices and communications in Canada are perceived to largely be informed by, and best serve, white Anglo-European settlers; a failure to challenge this practice is seen as a key barrier reducing effectiveness of these health protective measures for racially and ethnically minoritized Canadians. Overall, mainstream-aimed policies, practices and communication narratives commonly employed to protect Canadians were identified in this study as having a detrimental impact on rates of COVID-19 among South Asian communities, exacerbating stigma and discrimination towards these communities, ultimately re-iterating to underserved communities that they need to self-advocate for their own wellbeing. Given the intersectional implications of factors including race/ethnicity, income and job security as determinants of COVID-19 transmission, health officials have a critical role to play in addressing discrimination and stigma to promote equity-centered disease prevention efforts. This consideration should be integral to current and ongoing research and action related to pandemic preparedness.

## Figures and Tables

**Figure 1 ijerph-22-01668-f001:**
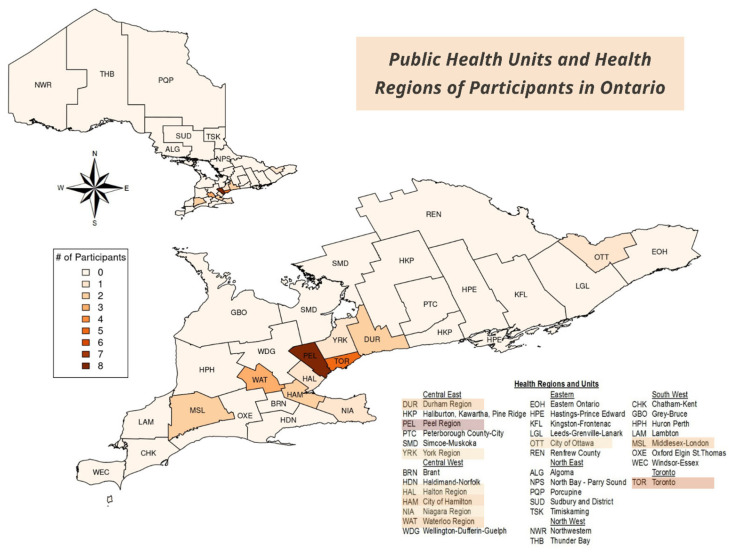
Map of Public Health Regions and Public Health Units across Ontario, Canada. Created using Public Health Ontario’s Easy Map Tool.

## Data Availability

Data are not available due to ethical restrictions.

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
