# Peer review of "Pushing for Structural Reforms: Impacts of Racism and Xenophobia upon the Health of South Asian Communities in Ontario, Canada"

_ijerph, 2025, doi:10.3390/ijerph22111668_

Round 1

Reviewer 1 Report

Comments and Suggestions for Authors

The argument of the article is clearly developed and there is full explanation of the research methodology, involving the conduct of three focus groups and the transcription and coding of the discussion.  It is noted that the analysis of the focus groups generated eight themes and 44 sub-themes, although they are not detailed in the article. There is reference to a second article that the authors or some authors co-wrote which discuss five of these themes (line 182); the remaining three themes are discussed in this paper.  There is detailed referencing of the literature, which will be useful to those wanting to extend their knowledge of the field.

The weakness of the article is the lack of detail.  For example, it is observed that there were “hot spots” in Ontario where there was disproportionate impact on South Asian communities of COVID-19 infection and mortality, without discussion of the actual impacts on different communities. There are footnote references, but no detail in the article.

In the article’s discussion of relative public health impacts on South Asian communities in Ontario there is little that is novel or unexpected – for example, the brief discussion of employment conditions and limited opportunities (crowded factories, “gig’ economy ...), multigenerational households, festivals, stigma, blaming, discrimination, media bias.

The substance of the argument is the (repetitive) discussion of Critical Race Theory. Primary research was hardly required as the argument largely operates at the theoretical level in its consideration of “the role of the structural determinants of health”:

“... structural analysis ... looks at how systems such as white supremacy, settler colonialism, capitalism, patriarchy and ableism work synergistically to manufacture social hierarchies which privilege some to the detriment of others” (line 456)

“Participants elaborated upon the compounding impacts of racism, income inequality, socioeconomic status and class upon the communities’ experience ...” (line 247)

Historical references to discrimination in 1914 and 1947 (p.2) do little to advance the argument, which would rather have benefitted from a focus on contemporary Canadian society.

With regard to the research that was conducted, the article fails to convey a sense that the authors were committed to a research design that would have the potential to challenge received wisdom. This point is raised because there were evident weaknesses in the achieved sample that were not addressed at the time of participant recruitment: thus, of the 25 participants in the three focus groups, most (18) were under the age of 29; 20 were women, and 84% had at least some university education. A further weakness was the advertising for participants only in English, although it is noted that the person conducting the focus groups had command of relevant languages.

In terms of what could have led to a better policy response to the pandemic by government and health authorities, given the urgent need, the focus on structural / embedded discrimination was of limited immediate utility as they could not be effectively addressed in a matter of weeks and months.  Initiatives to deal with immediate need warranted at least some discussion, to meet the objective of “research ...[that] provides insight into the role that health officials can play ... to promote equity-centred disease prevention efforts” in the context of a pandemic.  I was left wondering if anything was done well? In regions with which I am familiar, there were changes in service delivery, in government financial assistance, in communication strategies, and in the involvement of community leaders in program development.

Note: to enable blind review, the author’s names were removed in parts of the article (e.g. lines 202, 204, 211, 213, 217), but they are left in the heading on the first page (!!),in reference to an article written by some of the authors – footnote 22, line 183, and in the review report form.

Author Response

Many thanks for your detailed review. While we respectfully disagree with some of the concerns, we appreciate your time in reviewing our response in full. All changes to the manuscript are in red font. Thank you again for your time. 

Reviewer 2 Report

Comments and Suggestions for Authors

Overall, the paper is written well and presents important insights. There are some revisions that could improve the submitted work.

Introduction:

Given the readership of the IJERPH is international, the first para of the introduction section should have bring some global context (e.g. experiences for minority populations during the pandemic).

Methods:

  • Line 161: is the interviewer for focus groups a co-author of this paper?
  • Line 203: what is GTHA; please write in full
  • Line 211 and 215: What is PI; write in full
  • Line 215 to 218: unclear; please re-phrase

Results:

  • In the results section, there is a number provided after pseudonym of participants. Is that their study ID? I am not sure what is the purpose of providing Study ID. Often this is removed to enhance confidentiality of participants.
  • Line 326. The use of “Premier of Ontario” is pointing to a single person, but that fact is that it’s about political stance of the governing party. I suggest referring to something like “conservative leadership in Ontario at that time expressed sentiment that were…”

Discussion:

  • The Discussion section is too long. The first part (from line 365 to 444) presents the findings in too much detail, and it has led to redundancy between the Introduction/Results and Discussion. Please make this part short.
  • Line 406-408: something is missing in the sentence
  • Line 421: elaborate what was the issue about ‘vaccine prioritization’
  • Line 435-437: The discussion about ‘Apna Health’ should have been brought in the Result section, before discussing it in the Discussion section.

Typos:

Line 74: add ‘and’ between ‘Guo, Guo [13]’

Line 473: add ‘of’ between ‘cause systemic’

Line 487: ‘communicate’ should be replaced with ‘community’

Author Response

Many thanks for your detailed review. All changes to the manuscript are in red font. Thank you again for your time. 
